# Dnmt3a2 expression during embryonic development is required for phenotypic stability

Minmin Liu [1,4], Guillermo Urrutia[1,4], Rachel Shereda[1], Galen Hostetter[2], Stacey L. Thomas [1], Gangning Liang[3] & Peter A. Jones [1] ✉

Proper function and switching of regulatory elements are essential for vertebrate development and are regulated by DNA methylation. We use isoform-specific knockouts of the de novo methyltransferase Dnmt3a1 and Dnmt3a2 to probe their roles during embryogenesis and postnatal development. Mice lacking Dnmt3a1 show minimal embryonic methylation loss but are smaller and die postnatally. In contrast, $Dnmt3a2^{-/-}$ mice exhibit widespread hypomethylation at enhancers, CTCF sites and imprinted genes, which are largely repaired postnatally. These mice are viable but display sporadic abnormalities including anophthalmia, hydrocephalus, hydronephrosis and male infertility due to absence of sperm. Interestingly, the fertile $Dnmt3a2^{-/-}$ mice produce sperm with sporadic imprinting defects. These findings suggest that the two isoform have distinct, developmentally regulated roles, with Dnmt3a2 being crucial for maintaining proper methylation of regulatory elements, especially for enhancers, CTCF sites and imprinted genes, and preventing stochastic phenotypic outcomes after birth.

DNA methylation is an epigenetic modification that plays a critical role in regulation of gene expression by modulating the activity of key regulatory elements such as promoters, enhancers, CCCTC-binding factor (CTCF) binding sites, and Imprinting control regions (ICRs)[1]. These regulatory elements mediate spatial and temporal control of gene expression, ensuring precise transcriptional activation during development[2,3]. The control of gene expression is fundamental to proper formation and function of tissues and organs, as it dictates tissue-specific gene expression patterns[4,5]. However, there is still limited in vivo evidence linking impaired DNA methylation machinery to regulatory element dysfunction, particularly for enhancer and imprinted genes, which could ultimately lead to developmental abnormalities.

DNA methylation involves the addition of a methyl group to the 5′ position of cytosine residues within CpG dinucleotides, and is mainly established by the de novo DNA methyltransferases, including Dnmt3a and Dnmt3b, during development in mammals[6]. Mutations in *DNMT3A* have been implicated in a spectrum of human diseases, particularly hematological malignancies[7–11] and developmental disorders, including Tatton-Brown-Rahman syndrome (TBRS)[12] and microcephalic dwarfism[13]. These findings underscore the important role of *DNMT3A* in normal development and its contribution to disease when dysregulated.

Existing total and tissue-specific Dnmt3a KO mouse models have revealed diverse and context-dependent phenotypes, highlighting the importance of Dnmt3a in various biological processes. For example, homozygous Dnmt3a KO mice ($Dnmt3a^{-/-}$) display severe developmental defects, including growth retardation, premature death, and impaired germ cell development[6]. In contrast, heterozygous Dnmt3a KO mice ($Dnmt3a^{+/-}$) exhibit postnatal phenotypes such as obesity, increased bone length, and behavioral abnormalities, which partially recapitulate human TBRS caused by DNMT3A mutations[14,15]. Tissue-specific KO models further demonstrate that Dnmt3a is essential for germ cell development[16], hematopoietic stem cell (HSC) differentiation[17], neurogenesis[18,19], and suppression of tumorigenesis in a conditional mouse lung tumor model[20]. However, most studies investigating Dnmt3a function using mouse models have not distinguished between its two major isoforms, Dnmt3a1 and Dnmt3a2, which are generated through alternative promoter usage.

Dnmt3a1 features a 219-amino-acid-long N-terminal region, which is missing in Dnmt3a2, while the rest of the protein sequences are the same for both isoforms. The two isoforms exhibit distinct expression patterns, regulatory mechanisms, and potentially unique biological functions[21]. For example, while Dnmt3a1 is ubiquitously expressed and thought to be the predominant isoform in most somatic tissues, Dnmt3a2 is highly expressed

[1]Department of Epigenetics, Van Andel Institute, Grand Rapids, MI, USA. [2]Pathology and Biorepository Core, Van Andel Institute, Grand Rapids, MI, USA. [3]Department of Urology, Keck School of Medicine, University of Southern California, Los Angeles, CA, USA. [4]These authors contributed equally: Minmin Liu, Guillermo Urrutia. ✉e-mail: Peter.Jones@vai.org

in embryonic stem cells (ESCs) and during early development, suggesting it may have specialized roles in maintaining pluripotency or regulating rapid developmental transitions[21]. A recent study has reported isoform-specific mouse models and found that Dnmt3a1, but not Dnmt3a2, is essential for postnatal survival[22]. This highlights the role of Dnmt3a1 in the nervous system, regulating bivalent neurodevelopmental genes through interactions of its unique N-terminus with mono-ubiquitinated histone H2AK119[22]. In addition, Dnmt3a2 has been shown to be essential for maintaining genomic imprinting and epigenomic integrity in mouse embryonic cells[23–25]. These isoform-specific studies have enhanced our understanding of their distinct roles in development and disease. Despite their distinct expression profiles and potential functional differences, the individual contributions of these isoforms to methylate different regulatory elements during development remain poorly understood.

To dissect the specific roles of Dnmt3a1 and Dnmt3a2 in modulating the methylation of regulatory elements, we generated isoform-specific knockout models to investigate how each isoform contributes to the establishment and maintenance of methylation patterns during both embryonic and postnatal development. This approach provides an opportunity to unravel the different functions of these isoforms in modulating methylation of regulatory elements leading to eventually distinct developmental phenotypes.

## Results
### Dnmt3a2 knockout mice exhibit stochastic abnormalities during embryonic and postnatal development

To examine the function of the two different Dnmt3a isoforms during development, we generated isoform-specific *Dnmt3a* knock-out mouse

models. We used CRISPR-based genome editing to target exons 5 and 6 (744 bp) in mouse zygotes to delete *Dnmt3a1* specifically (Fig. 1A), resulting in frame shift and premature termination during translation (Supplementary Fig. 1). To delete Dnmt3a2, we targeted the promoter and exon1 of *Dnmt3a2* (941 bp), which are located within the intron 6 of the *Dnmt3a1* locus (Fig. 1A). Dnmt3a1 or Dnmt3a2 proteins were completely depleted in the respective E14.5 homozygous knockout embryos without affecting the expression of the other isoform (Fig. 1B, C). As previously reported[22], the growth of *Dnmt3a1*[−/−] mice was delayed, as shown by the reduced body weight compared to wildtype (WT) mice at postnatal day 21 (PD21) (Fig. 1D). All *Dnmt3a1*[−/−] pups were runted followed by death between 3 and 4 weeks after birth, showing similar phenotypes as the *Dnmt3a*-null mice[6,22]. The body weight of both male and female *Dnmt3a1*[+/−] mice exhibited a bimodal distribution, with increased variation compared to their WT littermates at PD21 (Fig. 1D), similar to previously reported phenotypes of *Dnmt3a*[+/−] mice[26]. The observation that Dnmt3a1 KO mice phenocopied Dnmt3a KO mice suggests that Dnmt3a1 is the predominant isoform and plays a critical role in postnatal development.

Although the male and female *Dnmt3a2*[−/−] mice were viable and their body weights showed no differences compared to their WT and *Dnmt3a2*[+/−] littermates at PD21 (Fig. 1E), sporadic *Dnmt3a2*[−/−] mice exhibited reduced (50%) bodyweight (4.5%, Fig. 1E). We also observed the incidence of sporadic developmental defects which were manifested from 14.5 days during embryonic development to 6 months postnatal (Fig. 2). These include hydronephrosis due to unilateral ureteral agenesis with a frequency of 5 out of 203 (2.5%) *Dnmt3a2*[−/−] mice (Fig. 2A and Table 1). The incidence of hydrocephalus was 2 out of 203 (1%) (Fig. 2B and Table 1), characterized by the entire brain being filled with cerebrospinal fluid. True anophthalmia

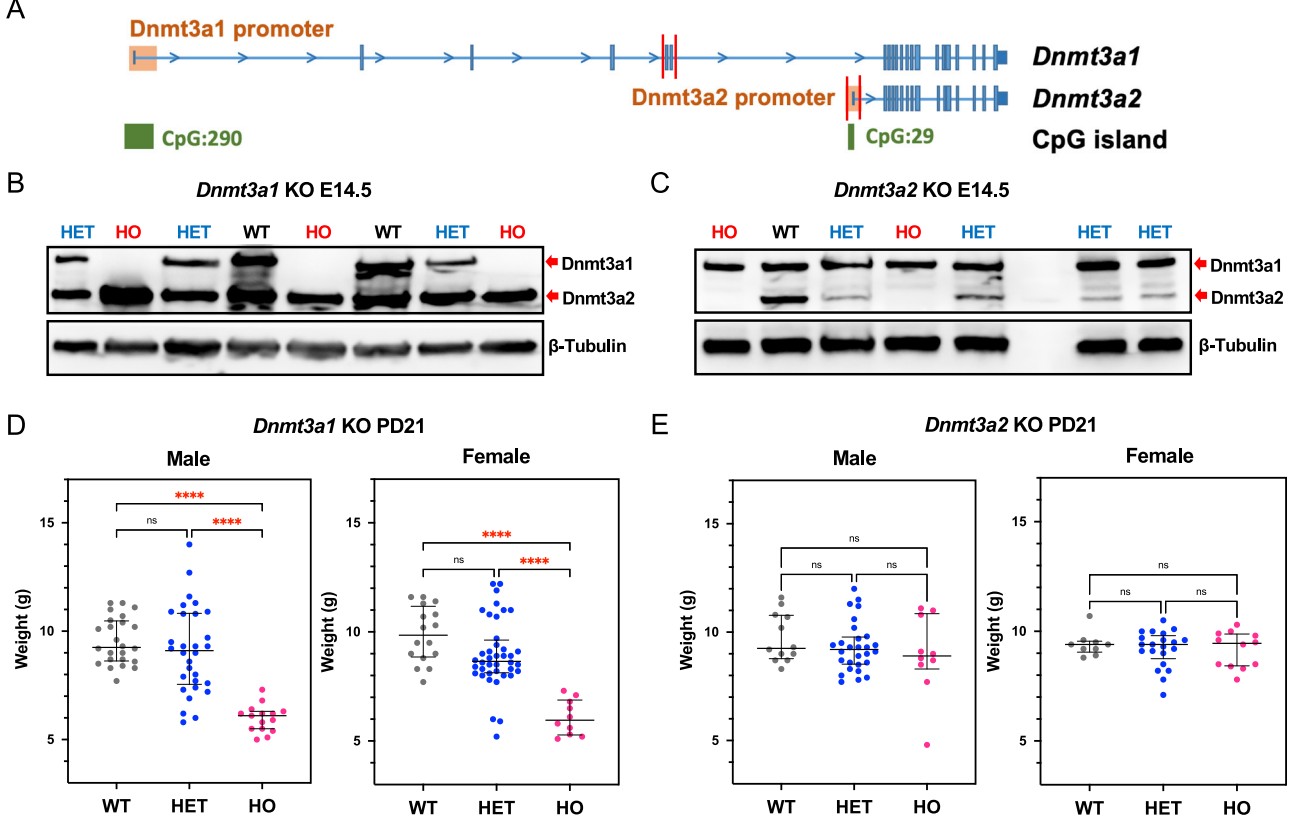

**Fig. 1 | Establishment of Dnmt3a isoform specific knockout mice. A** Diagram illustrating strategies by CRISPR-Cas9 genome editing to generate *Dnmt3a* isoform-specific KO models. The cut sites for each guide RNA were represented by red bars. Western blot analysis of E14.5 mouse embryos shows specific KO of Dnmt3a1 (**B**) and Dnmt3a2 (**C**) in homozygous (HO) embryos compared to their heterozygous (HET) and WT littermates. Body weights of *Dnmt3a1 KO* (**D**) and *Dnmt3a2 KO* (**E**) mice at 21 days postnatal. Biological replicates are represented by individual dots. Data shown as median ± 95% CI with individual values; ns not significant and ****, $p < 0.0001$ by one-way ANOVA with Tukey's multiple comparisons test.

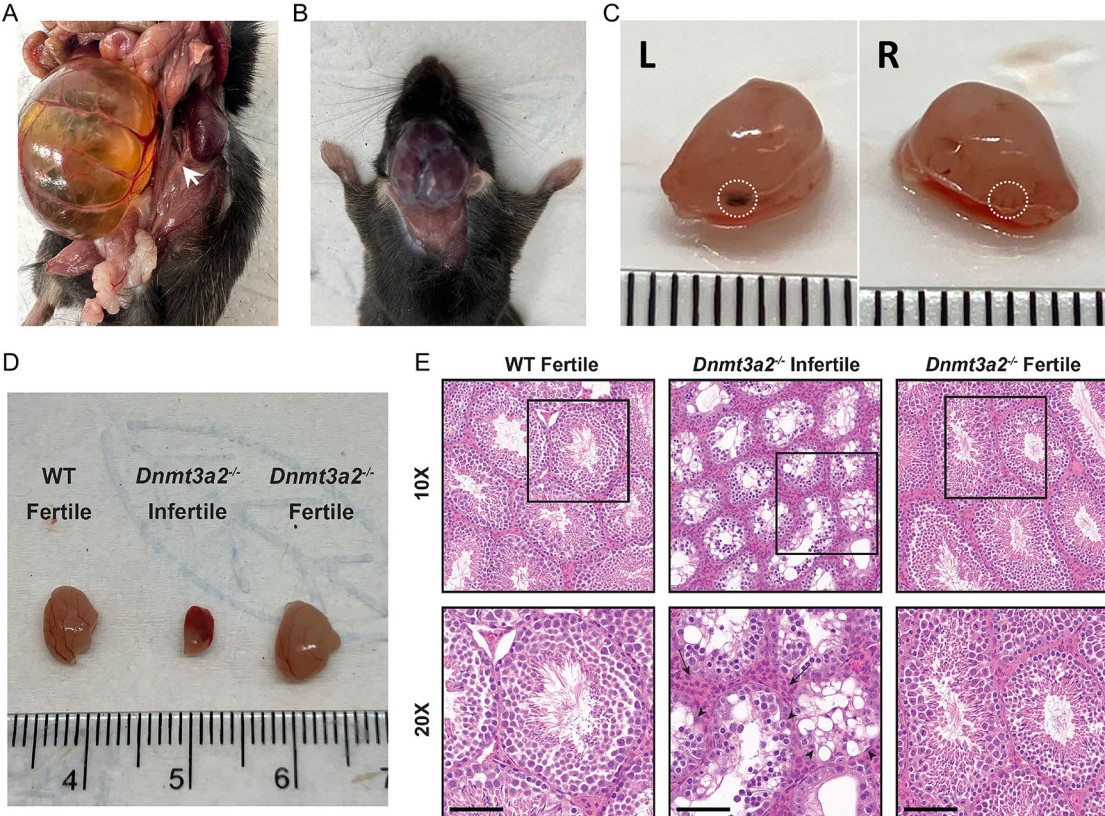

**Fig. 2 | Sporadic phenotypes in *Dnmt3a2*⁻/⁻ mice. A** Representative image of hydronephrosis in a five-month-old *Dnmt3a2*⁻/⁻ mouse. Note that there is no ureter connecting the right kidney with the bladder. White arrow shows normal ureter on the left side. **B** Representative image of hydrocephalus in a two-month-old *Dnmt3a2*⁻/⁻ mouse. **C** Representative image of anophthalmia in an E15.5 *Dnmt3a2*⁻/⁻ embryo. White dotted circle shows the hypoplastic eye on the right side. **D** Representative image comparing testicles from WT, Infertile, and Fertile *Dnmt3a2*⁻/⁻ mice. **E** Hematoxylin and eosin staining of WT, infertile, and fertile *Dnmt3a2*⁻/⁻ testicles. Pathology analysis reveals loss of spermatids, increase of interstitial cells (arrows), and reduced primary spermatogonia inside tubules (arrowheads), and remarkable complete loss of spermatozoa flagella in tubule lumens. Scale bar = 100 μm.

## Table 1 | Summary of the incidence of sporadic phenotypes observed in wildtype, *Dnmt3a1* and *Dnmt3a2* knockout mice

| Phenotypes | WT | Dnmt3a1⁺/⁻ | Dnmt3a2⁺/⁻ | Dnmt3a2⁻/⁻ |
|---|---|---|---|---|
| Hydronephrosis | 0/730 | 0/401 | 0/560 | 5/203 |
| Anophthalmia | 0/730 | 0/401 | 1/560 | 4/203 |
| Hydrocephalus | 0/730 | 0/401 | 0/560 | 2/203 |
| Male infertility | 0/82 | 0/89 | 0/69 | 4/70 |

also occurred in 4 out of 203 (2%) *Dnmt3a2*⁻/⁻ mice, and 1 out of 560 (0.18%) *Dnmt3a2*⁺/⁻ mice, characterized by the complete absence of one or both ocular structures due to failed optic vesicle formation during early embryogenesis (Fig. 2C and Table 1). In addition, out of 70 actively breeding males, 4 were identified as infertile, exhibiting an inability to successfully sire offspring despite confirmed mating activity, corresponding to an infertility rate of 5.7% within the tested cohort (Table 1). Analysis of a *Dnmt3a2*⁻/⁻ infertile mouse revealed absence of motile spermatozoa, providing a probable cause for its infertility. We isolated the testicles of the infertile *Dnmt3a2*⁻/⁻ mouse and compared to testes isolated from WT and fertile *Dnmt3a2*⁻/⁻ littermates. The testes isolated from the infertile animal were a third of the weight of the WT and fertile counterparts (Fig. 2D). Microscopic analysis of the infertile group showed testes with global hypoplasia with smooth glistening capsule with no evidence of mass or tumor (Fig. 2E). The seminiferous tubules were uniformly shrunken with noticeable loss of primary spermatogonia and lack of spermatocyte maturation. The stark

absence of spermatid production and lack of flagellum, and the tube centers being completely devoid of maturing spermatid. These microscopic features were noted in all fields of view and suggest a complete field-like phenotype. The uniformly diminished tubules were accompanied by diffuse increase of interstitial cells (of Leydig) but appear to be non-functional. Few primary and secondary spermatocytes were present, with irregular placement, indicating a disordered spermatocyte-spermatid maturation (Fig. 2E). Remarkably, in the fertile *Dnmt3a2*⁻/⁻ mice, careful microscopic review shows none of the features seen in the infertile group. Of note, the testes weight, spermatocyte maturation, spermatid formation and spermatozoa and distinctive flagellum were equivalent to the WT (Fig. 2E). Collectively, these stochastic defects affected a significant proportion (about 10%) of the *Dnmt3a2*⁻/⁻ mice, which were not found in the WT or *Dnmt3a1*⁺/⁻ mice (Table 1), although some developmental defects have previously been reported in the heavily inbred C57BL6 background (https://www. informatics.jax.org/inbred_strains/mouse/docs/C57BL.shtml). The observation that knocking out Dnmt3a2 increases the incidence of developmental defects in a stochastic manner suggests a role of Dnmt3a2 to ensure phenotypic stability during embryonic development.

### Dnmt3a2 participates in de novo methylation during embryonic development

We next investigated the DNA methylation changes that might be associated with the stochastic developmental defects observed in *Dnmt3a2*⁻/⁻ mice compared to their WT litter mates, focusing on two critical developmental windows: E15.5, and 21 days postnatal (PD21). These time points were selected to capture methylation dynamics during peak embryonic

DNA methylation establishment (E15.5) and during postnatal tissue maturation (PD21). To examine tissue-specific DNA methylation patterns, we isolated brain and liver tissues and used the Illumina MM285 Infinium Methylation EPIC array to perform genome-wide methylation analysis. Methylation values for each probe (CpG site) were expressed as beta values ($\beta$-value), defined as the ratio of methylated probe intensity to overall probe intensity[27]. We first examined the overall DNA methylation pattern in both male and female $Dnmt3a1^{-/-}$ and $Dnmt3a2^{-/-}$ mice compared to their WT littermates by visualizing the distribution of $\beta$-values for all autosomal probes. At E15.5, the median levels of DNA methylation for brain and liver in WT mice are 0.64 and 0.50, respectively, showing tissue-specific methylation patterns (Fig. 3A) (Supplementary Fig. 2A). While $Dnmt3a1^{-/-}$ embryos did not exhibit overall methylation changes (Fig. 3A) (Supplementary Fig. 2A), a significant reduction ($p < 0.001$) of the median DNA methylation levels were observed in $Dnmt3a2^{-/-}$ embryos (0.585 and 0.478 for brain and liver, respectively), showing hypomethylation in $Dnmt3a2^{-/-}$ embryos (Fig. 3B) (Supplementary Fig. 2B). These data indicate that Dnmt3a2 contributes to establish de novo methylation during embryonic development, while Dnmt3a1 is dispensable, despite being highly expressed at this stage (Fig. 1B). However, at PD21, substantial hypomethylation ($p < 0.001$) for $Dnmt3a1^{-/-}$ mice was observed (Fig. 3C) (Supplementary Fig. 2C), while the overall hypomethylation in $Dnmt3a2^{-/-}$ mice was not significant ($p = 0.0653$ for both brain and liver) (Fig. 3D) (Supplementary Fig. 2D). We also used a liquid chromatography-mass spectrometry (LCMS) method to measure the global 5mC level at PD21 and found a significant reduction ($p < 0.0001$) of 5mC level in the brains of $Dnmt3a1^{-/-}$ mice but not in the liver (Supplementary Fig. 3), possibly because site specific methylation patterns may not be captured at the global level.

To further quantify the level of hypomethylation, we used a 0.1 $\beta$-value difference and $p < 0.05$ as a cutoff to select the probes that showed consistent hypomethylation in our KO tissues compared to WT (Fig. 3E–H). In $Dnmt3a1^{-/-}$ mice, minimal hypomethylation was detected at E15.5 in both brain (0.041%) and liver (0.018%) (Fig. 3E and Table 2) (Supplementary Fig. 2E). However, by PD21, the level of hypomethylation escalated dramatically, reaching 26% in brain and 12% in liver, reflecting a profound postnatal decrease in the level of methylation (Fig. 3G and Table 2) (Supplementary Fig. 2G). This progressive lack of methylation suggests that Dnmt3a1 is dispensable during early embryonic development for de novo methylation but becomes critical for establishing methylation during postnatal maturation. Tissue-specific differences were pronounced, with brain exhibiting double the hypomethylation of liver, underscoring a heightened reliance on Dnmt3a1 in neural tissue for maintaining postnatal epigenetic stability as previously published[22].

In contrast, $Dnmt3a2^{-/-}$ mice displayed distinct embryonic-predominant hypomethylation. Applying the same $\beta$-value cutoff ($\geq$0.1 reduction vs. wildtype and $p < 0.05$), significant embryonic hypomethylation was observed at E15.5 in brain (8.1%) and liver (3.8%) (Fig. 3F and Table 2) (Supplementary Fig. 2F), consistent with the role of Dnmt3a2 in de novo methylation during organogenesis. Postnatally, hypomethylation levels declined to 1.3% in brain and 1.8% in liver by 21 days (Fig. 3H and Table 2) (Supplementary Fig. 2H), suggesting partial recovery of DNA methylation by Dnmt3a1 after birth. This temporal reversal implies that Dnmt3a2 is indispensable for establishing methylation patterns during embryogenesis, but its absence is partially mitigated postnatally, likely through compensatory mechanisms. Taken together, these findings suggest that Dnmt3a2 is mainly responsible for establishing de novo methylation during embryonic development (E15.5), while Dnmt3a1 plays the dominant role during the postnatal stage (PD21).

## Dnmt3a2 ensures proper de novo methylation especially at enhancers during embryonic development

As we mentioned previously, $Dnmt3a2^{-/-}$ mice exhibit stochastic phenotypes during embryonic and postnatal development. We then further investigated the genomic regions where the hypomethylated probes regulated by Dnmt3a2 are located, using classified genomic elements designed in

the MM285 array[27]. We then calculated the enrichment of hypomethylated probes in these genomic elements against their overall distribution in the mouse EPIC arrays. We found that the hypomethylated probes were highly enriched in enhancers, CTCF sites and monoallelic methylation regions in $Dnmt3a2^{-/-}$ brain and liver at E15.5 (Fig. 4A). Interestingly, the enrichment for enhancers disappeared mostly at PD21, but the CTCF sites and imprinted genes (MonoallelicMeth) were still enriched (Fig. 4A). In contrast, $Dnmt3a1^{-/-}$ brain and liver exhibit hypomethylation highly enriched exclusively for enhancers after birth (Fig. 4A). These results suggest that Dnmt3a2 is important for establishing DNA methylation at enhancers and CTCF binding sites during embryonic and/or postnatal development. In contrast, the observed enrichment of hypomethylation at imprinted genes (MonoallelicMeth) may result from defects in de novo methylation establishment in the gametes of the $Dnmt3a2^{/}$ parental mice. Although Dnmt3a2 is important to establish DNA methylation during embryonic development, its activity on enhancers could be fully compensated by the function of Dnmt3a1 after birth, but the activity of Dnmt3a2 on CTCF sites and imprinted genes cannot be compensated by Dnmt3a1 after birth.

As previously established, enhancers are tissue-specific, and our findings further demonstrate that Dnmt3a2 regulates enhancer DNA methylation in a tissue-specific manner. At E15.5, the enhancer related genes identified in brain and liver of the $Dnmt3a2^{-/-}$ mice were partially overlapped (Fig. 4B). We analyzed the gene ontology (GO) of enhancer-related genes using functional clustering of the GO terms and found that the overlapping genes in brain and liver were largely involved in multicellular organism development processes, while the other half of the hypomethylated enhancers in $Dnmt3a2^{-/-}$ embryos are near genes driving tissue specific development (Fig. 4B). Intriguingly, these methylation deficits were largely repaired after birth (Fig. 4C, D), possibly due to the compensatory activity of the remaining DNA methyltransferases, including the Dnmt3a1 isoform. The majority of probes remaining hypomethylated in PD21 $Dnmt3a2^{-/-}$ brain and liver were enriched for CTCF sites and imprinted genes (Fig. 4A). This tissue-specific methylation defect of development-related enhancers during embryonic development in $Dnmt3a2^{-/-}$ mice likely increases the chances of stochastic developmental defects observed in a subset of animals.

## Dnmt3a2 loss reduces the methylation of infertility-relate imprinted genes in sperm

Considering that male infertility was the most frequent phenotype in $Dnmt3a2^{-/-}$ mice (Table 1), and that the expression of this isoform remains present in adult mice testes[22], we hypothesized that the loss of Dntm3a2 might affect the DNA methylation profile in mouse sperm. For $Dnmt3a2^{-/-}$ mice, we failed to isolate any motile sperm from the one example we examined. We therefore isolated motile sperm from fertile WT and $Dnmt3a2^{-/-}$ mice and evaluated the DNA methylation patterns for all the autosomal probes using the EPIC array. Using a $\beta$-value cutoff of $\geq$0.2 and $p < 0.05$, we found that $Dnmt3a2^{-/-}$ sperm showed significant hypomethylation (6.5% of total) (Fig. 5A, B). The hypomethylated probes were enriched in four genomic elements: Monoallelic methylation (5.9%), PMD (4.0%), CTCF (1.6%), and Enhancer (1.4%) (Fig. 5C), indicating that Dnmt3a2 is required for complete DNA methylation of multiple regions in spermatozoa.

Imprinted genes are known to play a critical role in spermatogenesis, and alteration in their methylation can impede normal spermatogenesis[28–30]. Considering the importance of the activity of Dnmt3a2 on monoallelic methylation in the process of genomic imprinting (Fig. 4A), we further investigated if imprinted genes were hypomethylated in sperm from $Dnmt3a2^{-/-}$ mice. We identified 1497 hypomethylated probes distributed over 657 genes in the monoallelic methylation category (Fig. 5D). We compared the hypomethylated genes against a database of 150 previously described mouse imprinted genes and found 7 hypomethylated imprinted genes in $Dnmt3a2^{-/-}$ sperm: H19, Snrpn, Grb10, Ntm, H13, Zdbf2 and Rasgrf1 (Fig. 5D). These correspond to 40 hypomethylated probes located within imprinted differentially methylated regions (DMRs), including

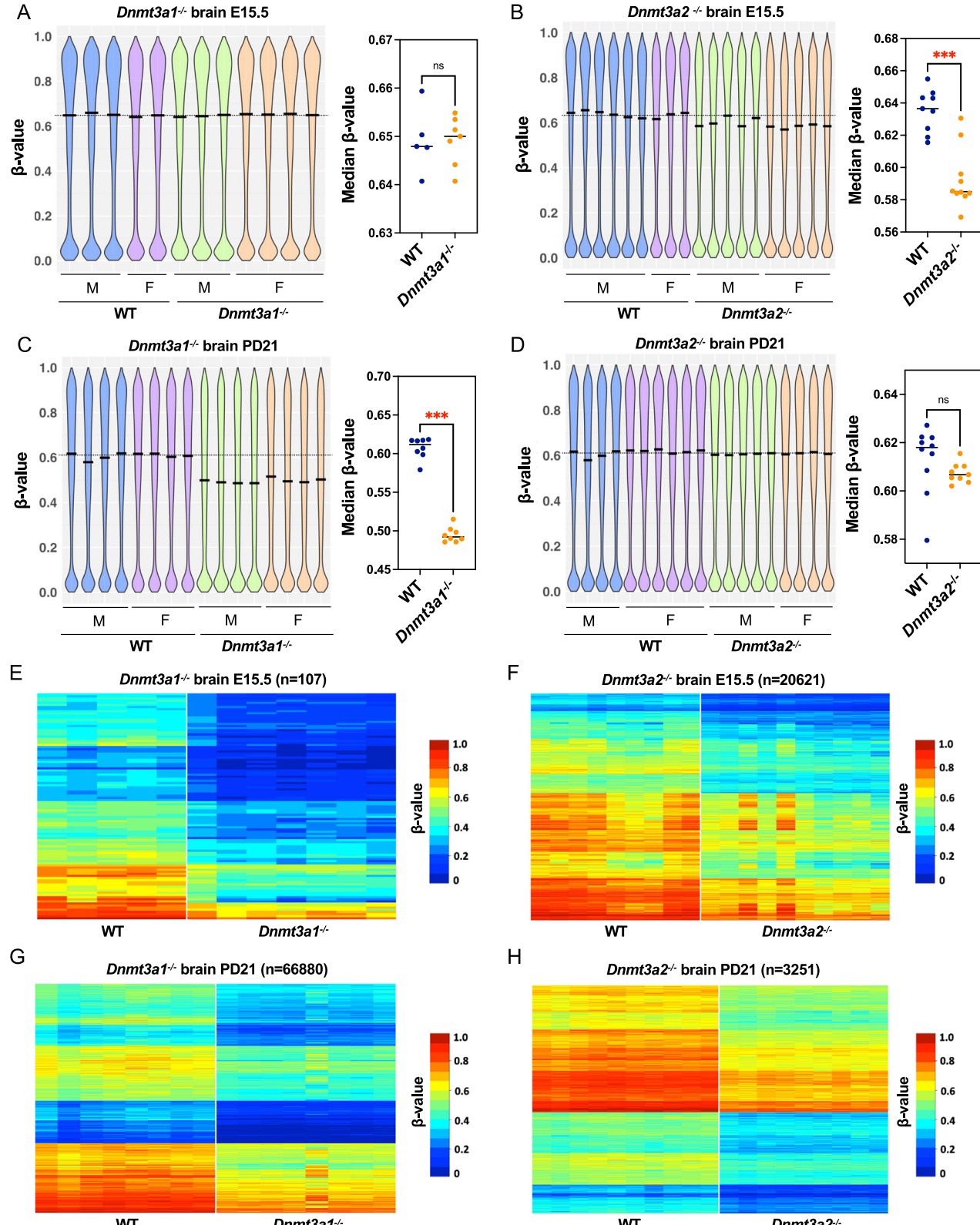

**Fig. 3 | Hypomethylated CpGs in brain from *Dnmt3a1*$^{-/-}$ and *Dnmt3a2*$^{-/-}$ mice during embryonic and postnatal development.** Distribution of CpG methylation for the autosomal probes from the MM285 array in brain of the *Dnmt3a1*$^{-/-}$ (**A, C**) and *Dnmt3a2*$^{-/-}$ (**B, D**) mice at E15.5 and PD21. The left panels show violin plots for the distribution of CpG methylation as *β*-values in WT and KO brain. Males are represented as M and females are represented as F. The median *β*-value in WT brain is marked by the dotted line. The right panels compare the median beta-value for the WT to the KO brain. Biological replicates are represented by individual dots. ns, not significant and ***, $p < 0.001$ by two-tailed unpaired Mann–Whitney U test. Heatmap representing the hypomethylated CpGs in brain of the *Dnmt3a1*$^{-/-}$ **E, G** and *Dnmt3a2*$^{-/-}$ **F, H** mice at E15.5 and PD21. Hypomethylated probes were selected using a cutoff of *β*-value difference greater than 0.1 and $p < 0.05$ by two-tailed unpaired Mann–Whitney U test. Methylation levels are represented by a cold to warm color scale (*β*-value 0–1, 0 – 100% methylated), where every row represents an individual probe, and every column represents sample from an individual mouse.

paternally methylated DMRs located on the *Gpr1-Zdbf2* domain (3 DMRs)[31], *H13*, and the *Rasgrf1* DMR[32] (Fig. 5E). These data indicate that *Dnmt3a2−/−* mice display a genomic imprinting defect in sperm. DNA methylation defects of *H19* and *Snrpn* have been previously associated with male infertility[30], suggesting that Dnmt3a2 expression may be necessary for proper methylation of imprinted genes associated with sperm development and fertility. However, most *Dnmt3a2−/−* mice remain fertile and produce sperm, while others are infertile and exhibit little to no sperm. This raises the question of whether proper methylation of imprinted genes is truly a driver of sperm development, or if infertility arises just due to stochastic abnormalities.

### Dnmt3a2 KO-induced methylation loss in sperm DNA follows a stochastic pattern

We further examined the hypomethylated regions of three imprinted genes using targeted amplicon bisulfite sequencing to include adjacent CpG sites not present in the array probes. For the *H19* gene, we confirmed that *Dnmt3a2−/−* sperm showed greater hypomethylation of the

CTCF1 (8%) compared to WT (2%), but not the CTCF2 binding site of the ICR (Fig. 6A, B and Supplementary Fig. 4). Similarly, the region located at the promoter of *H19* showed significant hypomethylation of the 4 CpG sites evaluated (Fig. 6A, C and Supplementary Fig. 5). In the *Snrpn*, and *Grb10*, each CpG site evaluated showed a different degree of hypomethylation compared to the controls. There was a 2-to-3-fold increase in the degree of hypomethylation for most sites (Fig. 6D, E and Supplementary Fig. 6–7). Additionally, we studied the *Snrpn* gene's paternally methylated region (DMR2, Supplementary Fig. 6), which spans exons 7–10, and has been previously described to be methylated during spermatogenesis[33]. Targeted amplicon sequencing of this region revealed hypomethylation of 1 of the 2 CpGs evaluated (Fig. 6D and Supplementary Fig. 6). Next, we evaluated the DNA methylation patterns of these regions in individual DNA sequences and observed that DNA methylation loss did not show a specific pattern (Supplementary Fig. 4-7). The number of unmethylated CpG sites per sequence was variable, and they appeared randomly distributed within the sequence. Additionally, our data shows that *Dnmt3a2−/−* mice exhibit DNA methylation defects to be stochastically distributed within the sperm population, as evidenced by the multiple unique methylation patterns shown in the DNA sequences evaluated (Supplementary Figs. 4–7). Previous reports have confirmed the expression of Dnmt3a2 isoform during the later stages of sperm maturation[34]. Our data once again indicated stochastic abnormalities, not only in infertile mice that do not produce mature sperm, but also in the loss of DNA methylation observed in sperm from fertile mice. This supports the idea that Dnmt3a2 expression is required to limit stochastic abnormalities during spermatogenesis, suggesting an active role for Dnmt3a2 at completing the methylation of multiple genomic regions involved in sperm development and maturation.

**Table 2 | Summary of the percentage of hypomethylated CpGs relative to all the CpGs represented in the mouse array in brain and liver from *Dnmt3a1−/−* and *Dnmt3a2−/−* mice**

| Hypomethylation % | E15.5 | PD 21 |
|---|---|---|
| *Dnmt3a1−/−* brain | 0.041% | 26% |
| *Dnmt3a1−/−* liver | 0.018% | 12% |
| *Dnmt3a2−/−* brain | 8.1% | 1.3% |
| *Dnmt3a2−/−* liver | 3.8% | 1.8% |

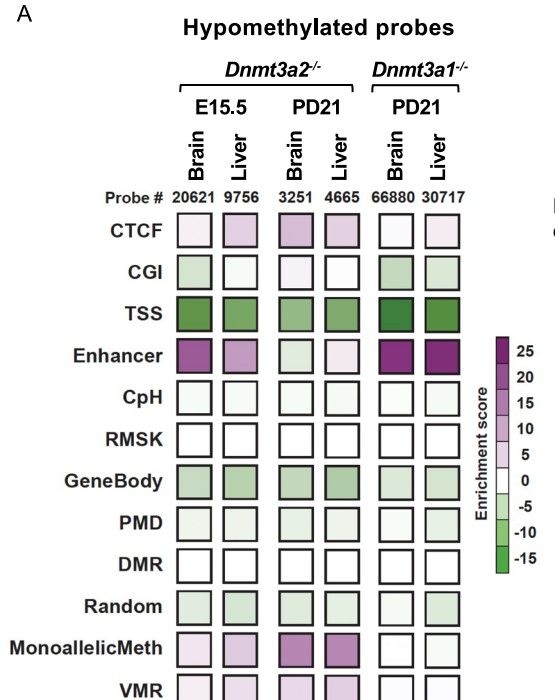

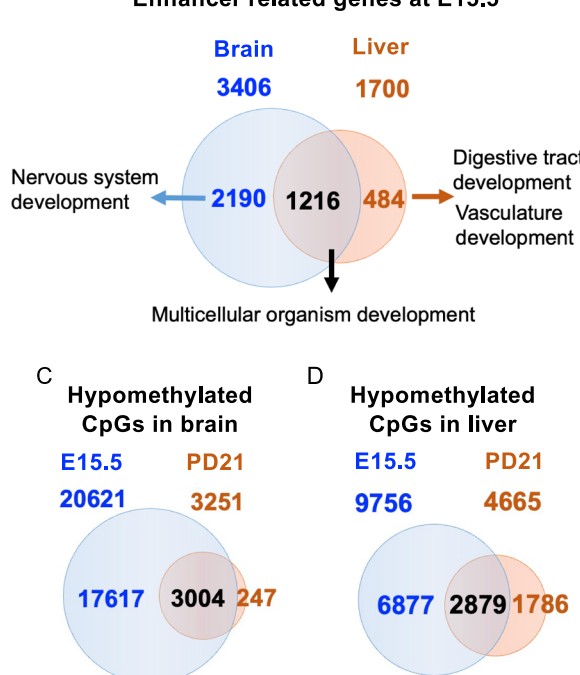

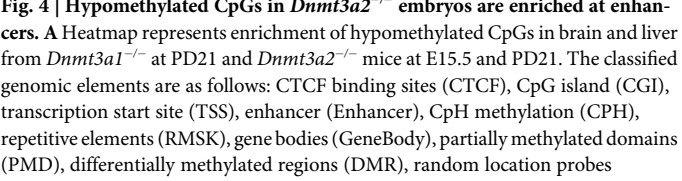

**Fig. 4 | Hypomethylated CpGs in *Dnmt3a2−/−* embryos are enriched at enhancers. A** Heatmap represents enrichment of hypomethylated CpGs in brain and liver from *Dnmt3a1−/−* at PD21 and *Dnmt3a2−/−* mice at E15.5 and PD21. The classified genomic elements are as follows: CTCF binding sites (CTCF), CpG island (CGI), transcription start site (TSS), enhancer (Enhancer), CpH methylation (CPH), repetitive elements (RMSK), gene bodies (GeneBody), partially methylated domains (PMD), differentially methylated regions (DMR), random location probes

(Random), monoallelic methylation, including the ICRs of imprinted genes (MonoallelicMeth), and metastable alleles and variably methylated regions (VMRs). **B** Venn diagram shows about half of the hypomethylated enhancers in *Dnmt3a2−/−* embryos are near genes driving tissue specific development. Venn diagrams show in brain (**C**) and liver (**D**), the methylation deficits in embryos were largely repaired after birth.

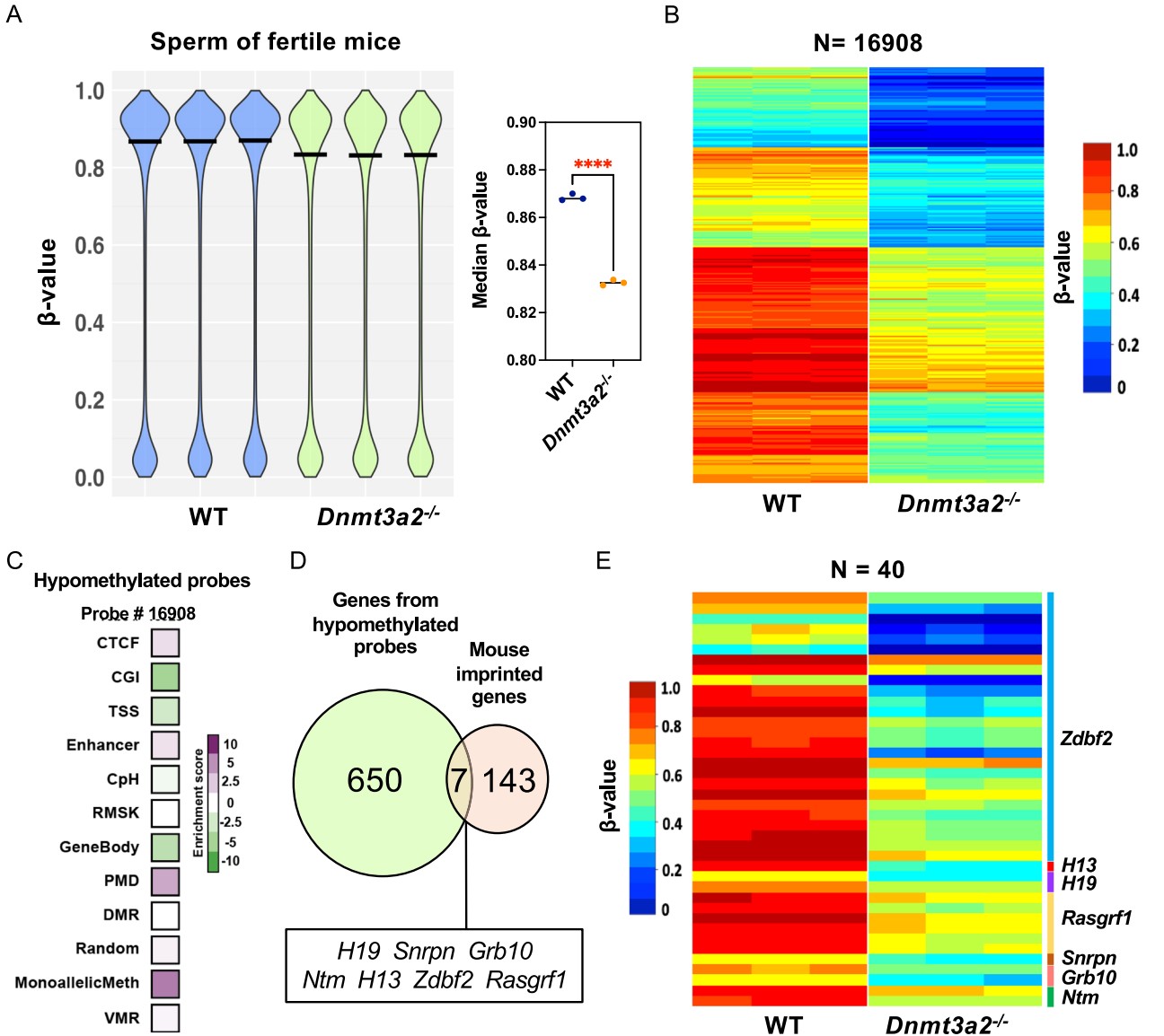

**Fig. 5 | Dnmt3a2 loss reduces the methylation of infertility-related imprinted genes in mouse sperm cells. A** DNA methylation violin plot showing the distribution of the β-values of the autosomal probes in the Infinium mouse EPIC array. β-values range from 0.0 to 1.0 for each probe, indicating total lack of methylation or complete methylation, respectively. The right panel compares the median β-value for the WT to the *Dnmt3a2*⁻/⁻ sperm. Biological replicates are represented by individual dots. ****, *p* < 0.0001 by two-tailed unpaired *t*-test. **B** Heatmap showing hypomethylated CpG sites in *Dnmt3a2*⁻/⁻ mice compared to WT. Hypomethylated probes were selected using a cutoff of β-value difference greater than 0.2 and *p* < 0.05 by two-tailed unpaired *t*-test. Methylation levels are represented by a cold to warm color scale (β-value 0-1, 0–100% methylated), where every row represents an individual probe, and every column represents sperm from an individual mouse. **C** Heatmap depicting the enrichment score of hypomethylated probes (DMPs) for multiple genomic elements included in the Infinium EPIC array. The classified genomic elements are as follows: CTCF binding sites (CTCF), CpG island (CGI), transcription start site (TSS), enhancer (Enhancer), CpH methylation (CPH), repetitive elements (RMSK), gene bodies (GeneBody), partially methylated domains (PMD), differentially methylated regions (DMR), random location probes (Random), monoallelic methylation, including the ICRs of imprinted genes (MonoallelicMeth), and metastable alleles and variably methylated regions (VMRs). **D** Venn diagram depicting the intersection between genes from monoallelic hypomethylated DMPs identified by the EPIC array and mouse imprinted genes. **E** Heatmap of hypomethylated CpG sites at imprinted DMRs in sperm from fertile *Dnmt3a2*⁻/⁻ mice. Probes were selected using a cutoff of a β-value difference > 0.2 and *p* < 0.05 (two-tailed unpaired *t*-test). Methylation levels (β-value from 0 to 1 representing 0–100% methylation) are indicated by a cold to warm color scale. Each row represents an individual probe, and each column represents an individual mouse.

## Discussion

Our findings highlight a developmentally regulated interplay between the Dnmt3a1 and Dnmt3a2 isoforms. Dnmt3a2 plays a central role in securing proper enhancer, CTCF sites, and imprinted gene methylation during embryogenesis, while Dnmt3a1 focuses on securing proper enhancer methylation in the postnatal stage of development. The compensation of methylation deficits after birth by Dnmt3a1 suggests a division of labor between the isoforms, with Dnmt3a2 acting as the primary de novo methyltransferase during early development and Dnmt3a1 completing

DNA methylation patterns postnatally. The increased incidence of sporadic phenotypes in Dnmt3a2-deficient mice underscores the potential importance of proper enhancer methylation in ensuring robust developmental outcomes. This study provides new insights into the isoform-specific functions of Dnmt3a and emphasizes the need to consider these distinct roles when investigating the mechanisms underlying functional elements, such as enhancers, CTCF sites, and ICRs, and developmental disorders.

The distinct functional roles of Dnmt3a isoforms during development cannot be fully explained by their different expression patterns alone. While

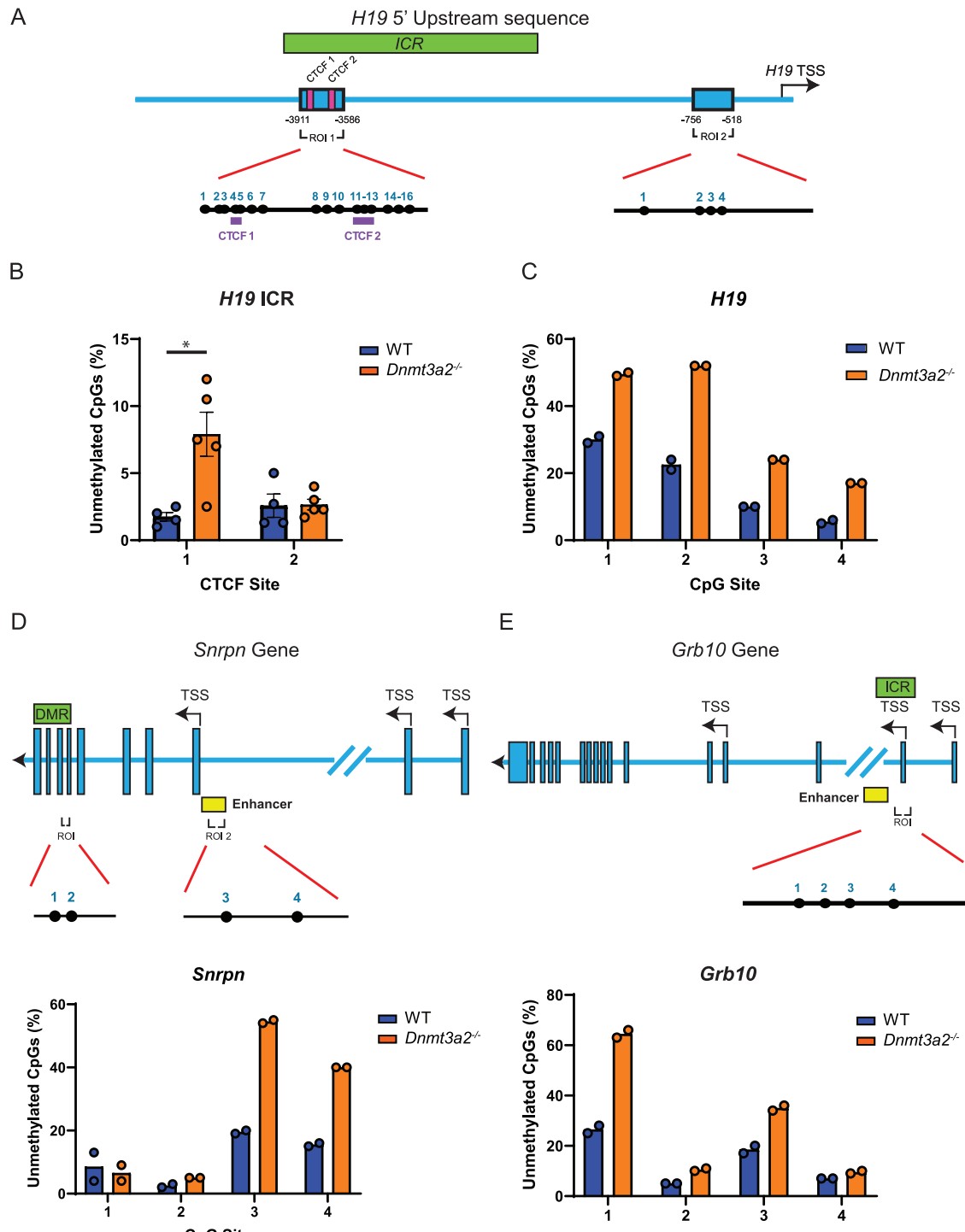

**Fig. 6 | Dnmt3a2 loss induces CTCF binding site loss of methylation at the *H19* ICR. A** Schematic representation of the *H19* 5' Upstream region. The studied regions of interest (ROI 1 and ROI 2, blue box) are located inside the ICR (Green box) encompassing two CTCF binding sites (magenta boxes), and at the promoter region of *H19*, respectively. Bar graphs depicting the percentage of unmethylated CpGs in sperm DNA from WT and *Dnmt3a2^−/−^* mice at the following loci: the *H19* ICR CTCF binding sites (ROI 1) (**B**), the *H19* promoter (ROI 2) (**C**), the *Snrpn* DMR and enhancer (**D**), and the ICR of *Grb10* (**E**). Schematic representation of the *Snrpn* and *Grb10* genes are shown in the top (**D**, **E**), where exons are depicted as blue boxes. The ICR/DMR is represented by a green box and the enhancer by a yellow box. The region of interest (ROI) amplified for targeted amplicon bisulfite sequencing is shown, with individual CpG sites (black lollipops) and their corresponding numbers indicated. Biological replicates are represented by individual dots. Data are presented as mean ± SEM. Significance was determined by two-tailed unpaired *t*-tests (*, *p* < 0.05).

our data demonstrate equivalent expression levels of Dnmt3a1 and Dnmt3a2 during embryogenesis in WT mice (Fig. 1B, C), their knockout models reveal different impacts on DNA methylation. Specifically, *Dnmt3a2^−/−^* embryos exhibit significant hypomethylation at E15.5, whereas *Dnmt3a1^−/−^* embryos show negligible methylation defects at this stage (Table 1). This suggests that Dnmt3a2 is the dominant isoform responsible for establishing de novo methylation during early embryonic development, likely through its collaboration with Dnmt3l to target enhancers and CTCF-

binding sites, as well as imprinted loci in the parental gametes[16,35,36]. In contrast, our data also demonstrates that Dnmt3a1 becomes indispensable postnatally, particularly at bivalent promoters and enhancers, where its loss provides the explanation of severe developmental abnormalities[22]. Our findings underscore a division of labor between the two Dnmt3a isoforms and highlight the subtle regulation of DNA methylation landscapes necessary for robust developmental outcomes.

The sporadic phenotypes observed in *Dnmt3a2*−/− mice further underscore the importance of methylation fidelity during embryogenesis. The incomplete penetrance of developmental defects in these mice suggests that Dnmt3a2-mediated methylation at enhancers acts as a buffer against intrinsic developmental heterogeneity—a phenomenon that is stochastic in nature[26,37,38]—thereby promoting robust developmental outcomes and limiting phenotypic variability. This is supported by our methylation analysis of spermatozoa from fertile *Dnmt3a2*−/− mice, which revealed stochastic hypomethylation at ICRs, promoters, and enhancers of imprinted genes compared to WT controls (Fig. 6 and Supplementary Fig. 4-7), potentially explaining the observed sporadic cases of male infertility[16,28–30]. It also supports the idea that Dnmt3a2 expression is required to minimize stochastic abnormalities during spermatogenesis, suggesting a potential active role for Dnmt3a2 in completing the methylation of multiple genomic regions involved in sperm development and maturation. While bulk methylation assays in E15.5 brain and liver tissues (using array-based approaches) reflect averaged methylation levels across heterogeneous cell populations (Fig. 3) (Supplementary Fig. 2), we can infer that Dnmt3a2 loss results in cell-to-cell variability of DNA methylation at enhancers during embryogenesis. Such stochastic hypomethylation likely disrupts the precision of transcriptional regulation in individual cells, ultimately manifesting as sporadic developmental phenotypes in *Dnmt3a2*−/− mice. Taken together, the loss of Dnmt3a2-mediated epigenetic regulation during embryogenesis appears to destabilize developmental processes, increasing the possibility of subsequent abnormalities.

The sporadic phenotypes observed in *Dnmt3a2*−/− mice, including anophthalmia, hydrocephalus, and hydronephrosis, which reflect profound developmental defects, such as the complete absence of the ureter (hydronephrosis), eye (anophthalmia), and potentially ciliary dysfunction (hydrocephalus), can be attributed to DNA methylation changes caused by the loss of Dnmt3a2 during critical developmental windows, rather than genetic mutations or environmental causes. For example, hydronephrosis in inbred C57BL/6 mice has been linked to autosomal recessive mutations in *Aqp2cph*, which induce nephrogenic diabetes insipidus and polyuria, leading to obstructive nephropathy without structural deficits in the pyeloureteral peristaltic machinery[39–41]. In such models, hydronephrosis arises postnatally due to functional urinary tract obstruction rather than developmental absence of the ureter. By contrast, the complete agenesis of the ureter in *Dnmt3a2*−/− mice implicates a failure in early developmental patterning, likely stemming from disrupted epigenetic regulation during critical morphogenetic windows. In addition, the incidence of hydrocephalus in *Dnmt3a2*−/− mice was 1% (2 out of 203), a 34-fold increase compared to the background level of 0.029% in the inbred C57BL/6 J strain (https://www.jax.org/news-and-insights/2003/july/hydrocephalus-in-laboratory-mice), strongly implicating Dnmt3a2 loss as a contributing factor. Similarly, the observed eye abnormalities, which may result from defects in lens development, and the severe hydronephrosis phenotype, which cannot be explained by genetic mutations alone, further support the role of epigenetic dysregulation in these defects. The fact that environmental factors like alcohol and other teratogens can exacerbate the rate of developmental defects suggests that epigenetic mechanisms, in addition to genetic factors, play a critical role in shaping developmental outcomes. Future studies mapping methylation dynamics at single-cell resolution in *Dnmt3a2*−/− embryonic tissues could directly test whether stochastic or uniform hypomethylation at lineage-specific enhancers underlies these phenotypes.

Our data also suggest that while the isoforms can compensate for each other's loss in certain contexts, they possess distinct, non-overlapping functions during critical developmental windows. While the hypomethylation in *Dnmt3a2*−/− embryos establishes it as the dominant isoform during embryogenesis, the viability of these embryos suggests its function can be partially compensated by Dnmt3a1 and/or Dnmt3b. Conversely, this compensatory capacity is insufficient postnatally, as evidenced by the lethality of *Dnmt3a1*−/− pups after weaning, indicating that Dnmt3a2 and/or Dnmt3b cannot compensate for the loss of the critical role of Dnmt3a1 in postnatal development[22]. This isoform-specific requirement is further highlighted in the male germline development. The canonical model of conditional Dnmt3a KO results in a complete meiotic arrest identical to *Dnmt3l*−/− mice, due to a failure to establish imprints[16,36]. In contrast, our *Dnmt3a2*−/− model exhibits incomplete penetrance of infertility, with only a subset of males showing late spermatocyte-spermatid maturation defects. This suggests that in the absence of Dnmt3a2, Dnmt3a1 and/or Dnmt3b can partially, but not always completely, rescue critical methylation events in the germline, which may result in phenotypic severity likely correlating with the degree of residual DNA methylation in these loci.

Lastly, our findings, together with other emerging evidence, suggest that the functional roles of Dnmt3a isoforms may extend beyond de novo methylation to include maintenance functions[42–47]. While we showed that Dnmt3a2 is important for establishing and maintaining genomic imprinting and epigenomic integrity in embryonic cells, previous studies have also implicated both Dnmt3a and Dnmt3b in the maintenance of DNA methylation at genomic regions prone to erosion, such as partially methylated domains and repetitive elements[42–44,46]. In addition, a recent study showed that DNMT3A interacts with UHRF1, providing a potential mechanism for its maintenance methylation activity[45]. Therefore, it is plausible that Dnmt3a2, and possibly Dnmt3a1, contribute to the stability of DNA methylation patterns not only during early development but also in somatic maintenance, ensuring long-term epigenetic fidelity.

In summary, this study provides new insights into the isoform-specific functions of Dnmt3a and highlights the importance of considering these distinct roles when investigating the epigenetic regulation of development and disease. The observed phenotypes in Dnmt3a2-deficient mice emphasize the potential consequences of enhancer dysregulation and suggest that Dnmt3a2 may serve as a safeguard against developmental instability. Future studies should explore the molecular mechanisms underlying the isoform-specific functions of Dnmt3a and their potential implications for human developmental disorders and diseases linked to epigenetic dysregulation. Overall, this work advances our understanding of the complex roles of Dnmt3a isoforms in shaping the epigenetic landscape and ensuring proper development.

## Methods

### Animals and husbandry

This research complies with ethical regulations, with protocols approved by the Institutional Animal Care and Use Committee (Van Andel Institute (VAI); protocols 20-06-006 and 23-06-013). We have complied with all relevant ethical regulations for animal use.

The *Dnmt3a1* and *Dnmt3a2* KO mice were generated by injecting CRISPR–Cas9/sgRNA mixtures into B6C3F2 zygotes and were performed by the VAI Transgenics Core (RRID:SCR_022914). Briefly, 50 ng/μl HiFi AltR Cas9 (Integrated DNA Technologies), 25 ng/μl crRNA:tracrRNA1 and 25 ng/μl crRNA:tracrRNA2 (Integrated DNA Technologies) mixed in modified Tris-EDTA (TE) buffer (10 mM Tris-HCl pH 7.4, 0.1 mM EDTA) were injected into B6C3F2 zygote cytoplasm. Sequences of gRNA are listed in Supplementary Table 1. Mosaic mice from injections were then back crossed to WT C57BL/6 J mice for at least seven generations to obtain heterozygous mice. The homozygous mice were generated by intercrossing the heterozygotes. Littermates were randomly assigned to cages, with a maximum of five mice per cage. Littermate controls were used for all comparisons between WT and mutants to control for potential confounders. For genotyping, ear punch biopsies were collected, and the primers used are listed in Supplementary Table 1. All mice were housed at VAI Vivarium core (RRID:SCR_023211) in individual ventilated cages

(Tecniplast, Sealsafe Plus GM500 in DGM Racks) with a 12-h/12-h light/ dark cycle at 22 ± 1 °C and 30%–70% humidity, and were fed breeder chow (LabDiet, 5021, 0006540). Mice were checked daily by animal keepers and two to three times per week by expert VAI Vivarium Core Staff who were blinded to the genotypes for well-being and any health concerns. Our mouse cohort includes animals screened for both early- and later-manifested defects, from 14.5 days during embryonic development to 6 months postnatal. To screen for early-manifested defects, timed matings were performed, and embryos were harvested at E14.5-E15.5, or pups were collected at 21 days postnatal. Mice with reported phenotypes were euthanized via CO2 asphyxiation.

## Tissue lysate preparation and western blot analysis
To examine Dnmt3a1 and Dnmt3a2 expression, brain tissue from E14.5 embryos was crushed and lysed in 200 µl T-PER™ Tissue Protein Extraction Reagent (Thermo Scientific) supplemented with 2% SDS and 1× protease inhibitors (Roche). The extracted proteins were denatured in SDS-loading buffer and analyzed by western blot using DNMT3A (E9P2F) Rabbit mAb (Cell Signaling Technology CST 49768) and β-tublin (D3U1W) Mouse mAb (Cell Signaling Technology CST-86298).

## Tissue preparation, DNA extraction, and pathology analysis
Brain (right cerebrum) and liver from E15.5 embryo and PD21 mice, as well as testes from adult mice (5 months old), were digested in lysis buffer (10 mM Tris-HCl pH 8.0, 10 mM EDTA, 1% SDS, and 2 mg/ml Proteinase K) overnight at 55 °C, followed by phenol-chloroform 1:1 isolation and 50% isopropyl alcohol precipitation. DNA pellet was then washed twice in 70% ethanol and dissolved in TE buffer (10 mM Tris-HCl, pH 8.0, 0.1 mM EDTA). Testes from WT and *Dnmt3a2*$^{-/-}$ adult mice were isolated and fixed uniformly in 10% NBF for 24 hours before histology processing, sectioning, and hematoxylin and eosin (H&E) staining by the histology department in Pathology Biorepository Core at VAI (RRID:SCR_022912). H&E-stained tissues were assessed and reviewed by the study pathologist.

## DNA isolation from fertile mice sperm
Cauda epididymis from WT and fertile *Dnmt3a2*$^{-/-}$ mice were removed and cleaned before suspending them in M16 buffer (95 mM NaCl; 4.8 mM KCl; 1.1 mM KH2PO4, 1.2 mM MgSO4, 0.15% Sodium lactate; 5.5 mM Glucose, 2.1 mg/ml NaHCO3, 0.36 mg/ml Sodium pyruvate, 2.3 mM CaCl$_2$, and 4 mg/ml BSA). Cauda epididymis was nicked, transferred to a tube, and incubated in M16 buffer for 2 h in an incubator (37 °C, 5% CO2) to allow healthy sperm to swim up. Supernatant containing sperm was removed without disrupting the bottom, and sperm spun down by centrifugation (3000 rpm, 6 min). Sperm was digested in lysis buffer (10 mM Tris-HCl, pH 8.0, 10 mM EDTA, 1% SDS, 0.1 M DTT, and 2 mg/ml Proteinase K) overnight at 55 °C, followed by phenol-chloroform 1:1 isolation and 100% ethanol precipitation. DNA pellet was then washed twice in 70% ethanol and dissolved in TE buffer (10 mM Tris-HCl, pH 8.0, 0.1 mM EDTA).

## Mouse DNA methylation array
DNA samples (20–500 ng) were bisulfite converted using the Zymo EZ DNA Methylation Kit (Zymo Research) following the manufacturer's protocol with modifications for the Illumina Infinium methylation assays. DNA methylation was assessed using Illumina Infinium Mouse Methylation BeadChip array (MM285), conducted by the VAI Genomics core (RRID:SCR_022913) following the manufacturer's specifications. Arrays were scanned on the Illumina iScan platform, and probe-specific calls were made using Illumina GenomeStudio version 2011.1 to generate IDAT files.

## DNA methylation analysis
Quality control, preprocessing, and normalization processes were performed using the SeSAMe pipeline (version 1.24.0) on Bioconductor[48], and the methylation status of individual CpG sites was reported as a β-value, ranging from 0 (unmethylated) to 1 (fully methylated). Probes located on

the sex chromosomes were excluded from the analysis to enable comparison between KO and WT tissues independent of sex. Probe enrichment analysis was performed using the SeSAMe knowYourCG module[48], with annotations based on the knowYourCG tool, and related information available at http://zwdzwd.github.io/InfiniumAnnotation#mouse. The ChromHMM annotation was derived from a mouse consensus by the ENCODE project profiling 66 mouse epigenomes across 12 tissues at daily intervals from embryonic day 11.5 to birth[49]. Gene ontology analysis of probe-enriched genes was performed using the Database for Annotation, Visualization, and Integrated Discovery (DAVID) functional annotation clustering (v2023q4)[50]. Further data visualization of SeSAMe output was performed using RStudio (2024.12.1.563) in R (version 4.4.2).

## DNA digestion to single nucleosides
Genomic DNA that had been isolated from mouse brain and liver tissue was further purified using Genomic DNA Clean & Concentrator-25 (Zymo Research, D4065) and eluted in 50 µL LC-MS grade water (Thermo Scientific, 51140). DNA was then digested to single nucleosides using Nucleoside Digestion Mix (New England Biolabs, M0649S). Each sample, containing 200–500 ng of DNA, was digested using 5 µl of Nucleoside Digestion Mix in a 40 µl reaction volume with 10X Reaction Buffer and LC-MS grade water. The reaction was incubated for 4 h at 37 °C and then stored at -80 °C until needed for mass spectrometry analysis. Some samples were used to verify digestion completeness by checking for the absence of ethidium bromide labeled intact DNA after the electrophoresis of 100 ng of digested DNA on an agarose gel.

## Nucleoside quantitation in DNA digests by LC-MS
Absolute nucleoside quantitation in DNA digests was accomplished using liquid chromatography (LC)-mass spectrometry (MS) in the Van Andel Institute Mass Spectrometry Core (RRID:SCR_024903). An external calibration curve was prepared from a stock mix containing 2′deoxycytidine (dC; 100 µg/mL) (30125, Caymen Chemical), 5-methyl-2′deoxycytidine (5mdC; 4 µg/mL) (16166, Caymen Chemical), and 2′deoxyguanosine (dG; 100 µg/mL) (9002864, Caymen Chemical). The stock mix underwent a 7-step serial dilution of half-log steps for eight total curve points. 35 µL of DNA digests and standard curve samples were extracted with 315 µL of 80% methanol (v/v) containing [13C515N]2′deoxycytidine (6.3 ng/mL) (D239552, Toronto Research Chemicals), [D3]5-methyl-2′deoxycytidine (12.6 ng/mL) (M295902, Toronto Research Chemicals), and [13C5] 2′deoxyguanosine (12.6 ng/mL) (D232617, Toronto Research Chemicals) as internal standards.

Samples and standards were analyzed with an Agilent 6470 triple quadrupole mass spectrometer coupled with an Agilent ultra-high performance liquid chromatography 1290 Infinity II. 2 µL of each sample was injected, separated using a 21-min gradient on a Cortecs T3 Column (1.6 µm, 2.1 × 150 mm, 186008500, Waters, Eschborn, Germany) combined with a Cortecs VanGuard cartridge (1.6 µm, 2.1 × 5 mm, 186008508, Waters). Mobile phase A consisted of LC/MS grade water (W6, Fisher) with 1 mM ammonium acetate (73594, Sigma), and 0.01% ammonium hydroxide (A470, Fisher). Mobile phase B consisted of 99% Acetonitrile (A955-4, Fisher) and 1% LC/MS grade water (W6, Fisher). Column temperature was kept at 30 °C, flow rate was held at 0.3 mL/min, and the chromatography gradient was as follows: 0–6 min held at 0% B, 6-9 min ramp from 0 to 10% B, 9-14 min ramp from 10% to 50% B, 14–18.9 min ramp from 50% to 99% B. At 19 min, flow is changed to 0% B at 0.4 ml/min, and held until 20.5 min, then decreased to 0.3 ml/min by 21 min. Mass spectrometer parameters were: Gas flow at 13 L/min at 80 °C, sheath gas flow at 11 L/min at 275 °C, and the nebulizer was set to 30 psi. Capillary voltage was +2500, and nozzle voltage was +500. Data were acquired using dynamic multiple reaction monitoring (dMRM), including at least two transitions per compound. The transition list was developed and optimized using neat analytical standards, and parameters are provided in Supplementary Table 2. The dMRM parameters were determined based on running LC/MS grade analytical standards for each target compound. Peak picking and integration were

performed using Skyline Software (v 24.1). The ISTD peak area was used to correct for matrix effect differences between the samples and standards. The corrected peak areas were used to generate a linear or quadratic regression for quantitation of the target analytes. The quantified analytes (ng/mL) were then used to calculate 5mdC/dG ratio.

### Targeted amplicon bisulfite sequencing and analysis

Bisulfite conversion of DNA was carried out using the EZ DNA Methylation-Gold Kit (Zymo Research, #D5006) according to manufacturer's recommendations. *H19* ICR region was amplified using a nested PCR approach. Briefly, 100 ng of bisulfite-converted DNA was amplified with primer set 1 using ZymoTaq Polymerase (Zymo Research, #E2002) under the following amplification conditions: 95 °C for 10 min, 35 cycles of amplification (95 °C for 30 s, 54 °C for 30 s, 72 °C for 40 seconds), final extension at 72 °C for 7 min. Second round of amplification was carried out using template from first reaction with the primer set 2 using same amplification conditions. *H19*, *Snrpn*, and *Grb10* regions were amplified using MyTaq Red Mix (Meridian Biosciences, BIO-25044) under the following amplification conditions: 95 °C for 2 min, 40 cycles of amplification (95 °C for 30 s, 58 °C for 30 s, 72 °C for 60 s), final extension at 72 °C for 5 min. Primers used are listed in Supplementary table 1. Bisulfite PCR products were sequenced using the Genewiz Amplicon-EZ sequencing service (Azenta Life Sciences) and aligned to reference sequences using the BISulfite-seq CUI Toolkit[51]. BAM files from sorted sequences were created using SAMtools[52] and loaded in IGV 2.17.4 for visualization and calculation of CpG dinucleotide methylation percentage.

### Statistical analysis

Biological replicates are represented by individual dots in the graphs. Experimental data were analyzed using Prism 10 software (GraphPad, version 10.4.1) by one-way ANOVA, two-tailed Student's *t*-test, or indicated in the corresponding figure legends. Statistical significance levels are denoted as follows: *, $p < 0.05$; **, $p < 0.01$; ***, $p < 0.001$; ****, $p < 0.0001$, ns, not significant. Data are presented as median ± 95% CI or mean ± SEM. as indicated in the corresponding figure legends. No statistical methods were used to predetermine sample size.

### Reporting summary

Further information on research design is available in the Nature Portfolio Reporting Summary linked to this article.

## Data availability

Data generated for this work have been deposited to GEO under accession number GSE295720. All source data behind the graphs can be found in the Supplementary Data, and the uncropped gels for western blots in figures are available in the Supplemenatry Information.

## Code availability

All publicly available codes and tools used to analyze the data are reported and referenced in the Methods. Any additional information required to reanalyze the data reported in this paper is available from the lead contact upon request.

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

## Acknowledgements

We gratefully acknowledge the support and services provided by the Core Technologies and Service at Van Andel Institute (VAI), including Vivarium (RRID:SCR_023211); Transgenics (RRID:SCR_022914); Pathology and Biorepository (RRID:SCR_022912); Mass Spectrometry (RRID:SCR_024903), and Genomics (RRID:SCR_022913) cores. This work was supported by the National Cancer Institute under awards R35CA209859 (P.A.J. and G.L.) and R50CA243878 (M.L.).

## Author contributions

M.L., G.L., and P.A.J. conceived the study. M.L., G.U., R.S., G.H., and S.L.T. performed experiments and analyzed the data. M.L. and G.U. wrote the manuscript with assistance from G.H., S.L.T., G.L., and P.A.J. P.A.J. supervised the study. M.L. and G.U. contributed equally to this work.

## Competing interests

P.A.J. is a consultant for Zymo Research Corporation. G.L. is a consultant for Pangea. All other authors declare no competing interests.
