## [Transparent Peer Review file · Communications Biology]

Dnmt3a2 expression during embryonic development is required for phenotypic stability

Corresponding Author: Dr Peter Jones

Version 0:

Reviewer comments:

Reviewer #1

(Remarks to the Author)

In this manuscript, Liu et al. investigate the roles of Dnmt3a1 and Dnmt3a2 in genome-wide methylation regulation during development. While the phenotypes of Dnmt3a1^{-/-} mice had been previously characterized, the key finding here is that Dnmt3a2-deficient mice displayed sporadic abnormalities at a low frequency, including male infertility which is likely due to hypomethylation of imprinted genes (e.g., H19) in sperm. This work underscores the distinct regulatory functions of Dnmt3a isoforms during embryonic and postnatal development. However, several aspects of the manuscript require further improvement:

1. The author stated that the cases for the stochastic developmental defects were over 15% in Dnmt3a2^{-/-} mice. This proportion (15%) was calculated inaccurately. Clearly the percentages of 4.5% (reduced body-weight) and 5.7% (infertility) were obtained using smaller cohorts/denominators.

It seems that the incidence of hydronephrosis, hydrocephalus and anophthalmia was analyzed within the same group of Dnmt3a2^{-/-} mice (n=203). However, anophthalmia was observed during early embryogenesis (at least for some cases), how did the other developmental defects were evaluated for these mice?

2. The infertile Dnmt3a2^{-/-} mice cannot produce motile sperm, therefore the authors isolated motile sperm from other fertile mice for DNA methylation analysis. Based on the characterization of Dnmt3a conditional mutant males (Nature 2004), there should be plenty spermatogonia in the seminiferous tubules around PD21. Therefore, histological examinations of the testes from infertile Dnmt3a2^{-/-} mice are required. Also, the imprinting status of H19 in isolated spermatogonia from the infertile Dnmt3a2^{-/-} mice should be analyzed.

3. The analyzed Snrpn enhancer region (only two CpG sites) is not the typical imprinting control region, which is located at the promoter of Snrpn and is usually not methylated in male germ cells.

4. Some errors need to correct:

- a. The numbers of genes shown in the Fig.4d should be 654-4-146.
- b. Line 271, delete 'change'?
- c. Use unified format for figure citation: Fig. 4a, b; Fig. 5a, 5b; Fig. S4 to S7 etc. And in line 272, "Fig. 5d, 5d" should be "Fig. 5d, e".

Reviewer #3

(Remarks to the Author)

In this study, the authors generated knockout mice targeting two specific isoforms of Dnmt3a, i.e. Dnmt3a1 and Dnmt3a2. They found that these two isoforms may have some distinctive functions in embryonic development and postnatal development. Dnmt3a1 mutant embryos showed minimal loss of DNA methylation, whereas Dnmt3a2 mutant embryos displayed hypomethylation at enhancers, CTCF sites as well as a few imprinted genes. In contrast, Dnmt3a1 mutant mice were smaller and died about 4 weeks after birth, which were probably caused by postnatal demethylation. Dnmt3a2 mutant

mice were viable, with sporadic abnormalities. These are interesting results with implications for their distinct roles in embryonic and postnatal development.

There are a few points that may be addressed if possible. I think these may make this interesting study more informative.

Main points:

1. Although the isoform-specific functions are interesting, it will be great if the authors also discuss about the redundant functions of these two Dnmt3a isoforms in embryonic and postnatal development. This may be done through comparisons of these isoform-specific knockout mice with the Dnmt3a knockout mice.

2. There are a few recent studies as well as some previous studies indicating the functions of Dnmt3a as well as Dnmt3b in maintenance DNA methylation. Also the authors mentioned in Line 79-81 of the manuscript that Dnmt3a2 was proven to be essential for maintaining genomic imprinting and epigenomic integrity in mouse embryonic cells (ref. 23-25). Therefore, it is reasonable to argue that Dnmt3a2 as well as Dnmt3a1 may be involved in maintenance DNA methylation, not just de novo DNA methylation. Is it worth discussing about this possibility?

3. Line 156-158, Dnmt3a2 contributes to de novo methylation during embryonic development, while Dnmt3a1 is dispensable, despite being highly expressed (Fig. 1b). Were they partially redundant? Can the authors discuss about it in the manuscript?

4. In Fig. 5, hypomethylation was found at one H19 ICR and three imprinted genes (H19, Snrpn, Grb10). How about other ICRs? It will be good to show DNA methylation results lacking Dnmt3a1 or Dnmt3a2 for most if not all of more than 20 known ICRs in mice.

Minor points:

1. Some Dnmt3a may need to be either italic or capital letters.

2. It might be better to use "genomic imprinting" than "gene imprinting" in a few places. For example, Line 256.

Version 1:

Reviewer comments:

Reviewer #1

(Remarks to the Author)

The revised manuscript demonstrates significant improvement through the inclusion of additional experiments and expanded discussion. While the DNA methylation analysis of germ cells from infertile Dnmt3a2^{-/-} males remains unresolved, I believe the current version is now suitable for publication in Communications Biology.

Reviewer #3

(Remarks to the Author)

The authors have properly addressed the concerns.

Reviewers' comments:

We sincerely thank the reviewers for their thoughtful review and for recognizing the significance of our findings regarding the distinct roles of the Dnmt3a isoforms. We also appreciate their constructive comments, which have helped us significantly improve our manuscript. We have performed several additional experiments to reinforce the conclusions and addressed each of the points raised in the sections below, and we believe the revisions have strengthened our study.

Reviewer #1 (Remarks to the Author):

In this manuscript, Liu et al. investigate the roles of Dnmt3a1 and Dnmt3a2 in genome-wide methylation regulation during development. While the phenotypes of Dnmt3a1^{-/-} mice had been previously characterized, the key finding here is that Dnmt3a2-deficient mice displayed sporadic abnormalities at a low frequency, including male infertility which is likely due to hypomethylation of imprinted genes (e.g., H19) in sperm. This work underscores the distinct regulatory functions of Dnmt3a isoforms during embryonic and postnatal development. However, several aspects of the manuscript require further improvement:

We thank the reviewer for the positive appraisal of our key findings and constructive critique.

1. The author stated that the cases for the stochastic developmental defects were over 15% in Dnmt3a2^{-/-} mice. This proportion (15%) was calculated inaccurately. Clearly the percentages of 4.5% (reduced body weight) and 5.7% (infertility) were obtained using smaller cohorts/denominators.

It seems that the incidence of hydronephrosis, hydrocephalus and anophthalmia was analyzed within the same group of Dnmt3a2^{-/-} mice (n=203). However, anophthalmia was observed during early embryogenesis (at least for some cases), how did the other developmental defects were evaluated for these mice?

This is an excellent point raised by the reviewer. First, we have removed the 15% claim which is mathematically incorrect as the reviewer pointed out. We therefore changed the conclusion to: "Collectively, these stochastic defects affected a significant proportion (about 10%) of the Dnmt3a2^{-/-} mice, which were not found in the WT or Dnmt3a1^{+/-} mice (Table 1), although some

developmental defects have previously been reported in the heavily inbred C57BL6 background.” (lines 141-145 in the revised manuscript).

We apologize for the lack of clarity in our original description. Our mouse cohort includes animals screened for both early and later-manifested defects, including the embryos harvested around E14.5-E15.5 and pups at PD21 which were used for further analysis in our manuscript. The single case of anophthalmia identified at E15.5 was from a litter that was included in the overall cohort count. The phenotypes for the vast majority of mice (including the other three cases of anophthalmia) were identified postnatally through daily health checks conducted by vivarium staff who were blinded to the genotypes. Notably, the observed congenital defects were mutually exclusive; for instance, a mouse presenting with anophthalmia was not also found to have hydrocephalus or hydronephrosis, preventing double-counting. This standardized protocol ensured a consistent assessment for all listed defects throughout the study period. We now provide a more clear and detailed description in the methods section of the revised manuscript (lines 569-575).

2. The infertile *Dnmt3a2*^{-/-} mice cannot produce motile sperm, therefore the authors isolated motile sperm from other fertile mice for DNA methylation analysis. Based on the characterization of *Dnmt3a* conditional mutant males (Nature 2004), there should be plenty spermatogonia in the seminiferous tubules around PD21. Therefore, histological examinations of the testes from infertile *Dnmt3a2*^{-/-} mice are required. Also, the imprinting status of H19 in isolated spermatogonia from the infertile *Dnmt3a2*^{-/-} mice should be analyzed.

We thank the reviewer for this insightful and important comment. In response, we have now performed detailed histological examinations of testes from an infertile *Dnmt3a2*^{-/-} mouse (5 months old) with WT and fertile *Dnmt3a2*^{-/-} littermates as controls. Interestingly, our analysis reveals a distinct phenotype that differs from the complete spermatogenic arrest previously reported in *Dnmt3a* conditional KO and *Dnmt3l*^{-/-} models. Instead, the infertile *Dnmt3a2*^{-/-} testes exhibit a specific defect in spermatocyte-spermatid maturation, characterized by the disorganized presence of aberrant primary and secondary spermatocytes. These new data are now included in the Results section (lines 124-141), discussed in the manuscript (lines 400-408), and presented as new Fig. 2D, E (see the figure presented below).

D

E

Please note, most *Dnmt3a2*^{-/-} mice remain fertile and produce sperm, while the infertile mouse exhibit little to no sperm. This raises the question of whether proper methylation of imprinted genes is truly a driver of sperm development, or if infertility arises just due to stochastic abnormalities (see revised manuscript lines 277-280). Also, the reviewer suggested that we analyze the imprinting status of H19 in isolated spermatogonia from the infertile *Dnmt3a2*^{-/-} mice. We agree that this would be a valuable experiment. However, due to the very low and unpredictable penetrance of the infertile phenotype in our model (only ~6% of males, 4 out of 70), it is very difficult to obtain infertile *Dnmt3a2*^{-/-} mice for isolating spermatogonia and perform bisulfite analysis. We believe that the technical challenges presented by the low penetrance of infertility make this experiment unfeasible with our current model and resources.

3. The analyzed *Snrpn* enhancer region (only two CpG sites) is not the typical imprinting control region, which is located at the promoter of *Snrpn* and is usually not methylated in male germ cells. We agree with the reviewer that the analyzed enhancer region of *Snrpn*, containing only two CpG sites, is not the typical paternally unmethylated imprinting control region (ICR). In addition to this region, we specifically investigated the paternally methylated region of the *Snrpn* gene (DMR2, spanning exons 7-10),

which is known to acquire methylation during spermatogenesis. Subsequent targeted bisulfite sequencing of this area revealed significant hypomethylation at 1 of the 2 CpG sites analyzed. This hypomethylation at a paternally methylated locus provides further evidence for a defect in the fulfillment of paternal imprints in the fertile *Dnmt3a2*^{-/-} sperm. We have summarized these results as new Fig. 6D (see the figure on the right), the new Supplementary Fig. 6 (see the figure below), and in the Result section (lines 289-296) of the revised manuscript.

4. Some errors need to correct:

a. The numbers of genes shown in the Fig.4d should be 654-4-146.

b. Line 271, delete 'change'?

c. Use unified format for figure citation: Fig. 4a, b; Fig. 5a, 5b; Fig. S4 to S7 etc.

And in line 272, "Fig. 5d, 5d" should be "Fig. 5d, e".

We thank the reviewer for identifying these errors. We have corrected them as follows:

a. The number of genes in Fig. 5D has been corrected (see the figure below). In addition, we found 33 probes that map to 3 paternal imprinted genes (H13, Zdbf2, and Rasgrf1). We therefore included this data in our revised Fig. 5D and as new Fig. 5E.

b. We have deleted the word "change" and modified this sentence: "There was a 2-to-3-fold increase in the degree of hypomethylation for most sites (Fig. 6D, E) (Supplementary Fig. 6-7)." (lines 289-292 of the revised manuscript).

c. We have corrected the mistake and unified the format for all figure citations throughout the manuscript to the consistent style.

Reviewer #3 (Remarks to the Author):

In this study, the authors generated knockout mice targeting two specific isoforms of Dnmt3a, i.e. Dnmt3a1 and Dnmt3a2. They found that these two isoforms may have some distinctive functions in embryonic development and postnatal development. Dnmt3a1 mutant embryos showed minimal loss of DNA methylation, whereas Dnmt3a2 mutant embryos displayed hypomethylation at enhancers, CTCF sites as well as a few imprinted genes. In contrast, Dnmt3a1 mutant mice were smaller and died about 4 weeks after birth, which were probably caused by postnatal demethylation. Dnmt3a2 mutant mice were viable, with sporadic abnormalities. These are interesting results with implications for their distinct roles in embryonic and postnatal development. There are a few points that may be addressed if possible. I think these may make this interesting study more informative.

We appreciate the reviewer's recognition of the interest and implications of our findings.

Main points:

1. Although the isoform-specific functions are interesting, it will be great if the authors also discuss about the redundant functions of these two Dnmt3a isoforms in embryonic and postnatal development. This may be done through comparisons of these isoform-specific knockout mice with the Dnmt3a knockout mice.

We are grateful to the reviewer for this excellent and insightful suggestion. In response, we have now added a new discussion section on redundant functions of the Dnmt3a isoforms to the revised manuscript (lines 393-409) as: "Our data also suggest that while the isoforms can compensate for each other's loss in certain contexts, they possess distinct, non-overlapping functions during critical developmental windows. While the hypomethylation in *Dnmt3a2*^{-/-} embryos establishes it as the dominant isoform during embryogenesis, the viability of these embryos suggests its function can be partially compensated by Dnmt3a1 and/or Dnmt3b. Conversely, this compensatory capacity is insufficient postnatally, as evidenced by the lethality of *Dnmt3a1*^{-/-} pups after weaning, indicating that Dnmt3a2 and/or Dnmt3b cannot compensate for the loss of the critical role of Dnmt3a1 in postnatal development²². This isoform-specific requirement is further highlighted in the male germline development. The canonical model of conditional Dnmt3a KO results in a complete meiotic arrest identical to *Dnmt3l*^{-/-} mice, due to a

failure to establish imprints^{16,36}. In contrast, our *Dnmt3a2*^{-/-} model exhibits incomplete penetrance of infertility, with only a subset of males showing late spermatocyte-spermatid maturation defects. This suggests that in the absence of Dnmt3a2, Dnmt3a1 and/or Dnmt3b can partially, but not always completely rescue critical methylation events in the germline, which may result in phenotypic severity likely correlating with the degree of residual DNA methylation in these loci.”

2. There are a few recent studies as well as some previous studies indicating the functions of Dnmt3a as well as Dnmt3b in maintenance DNA methylation. Also the authors mentioned in Line 79-81 of the manuscript that Dnmt3a2 was proven to be essential for maintaining genomic imprinting and epigenomic integrity in mouse embryonic cells (ref. 23-25). Therefore, it is reasonable to argue that Dnmt3a2 as well as Dnmt3a1 may be involved in maintenance DNA methylation, not just de novo DNA methylation. Is it worth discussing about this possibility?

We thank the reviewer for raising this key point regarding the potential role of Dnmt3a isoforms in maintenance methylation. We have now incorporated a discussion of this possibility into the revised manuscript (lines 411-421) as: “Lastly, our findings, together with other emerging evidence, suggest that the functional roles of Dnmt3a isoforms may extend beyond *de novo* methylation to include maintenance functions⁴²⁻⁴⁷. While we showed that Dnmt3a2 is important for establishing and maintaining genomic imprinting and epigenomic integrity in embryonic cells, previous studies have also implicated both Dnmt3a and Dnmt3b in the maintenance of DNA methylation at genomic regions prone to erosion, such as partially methylated domains and repetitive elements^{42-44,46}. In addition, a recent study showed that DNMT3A interacts with UHRF1, providing a potential mechanism for its maintenance methylation activity⁴⁵. Therefore, it is plausible that Dnmt3a2, and possibly Dnmt3a1, contribute to the stability of DNA methylation patterns not only during early development but also in somatic maintenance, ensuring long-term epigenetic fidelity.”

3. Line 156-158, Dnmt3a2 contributes to de novo methylation during embryonic development, while Dnmt3a1 is dispensable, despite being highly expressed (Fig. 1b). Were they partially redundant? Can the authors discuss about it in the manuscript?

We thank the reviewer for raising the crucial point regarding the potential for partial functional redundancy between Dnmt3a1 and Dnmt3a2 during embryogenesis. Together with the

response to comment#1, we now include these discussions in the revised manuscript (lines 393-409).

4. In Fig. 5, hypomethylation was found at one H19 ICR and three imprinted genes (H19, Snrpn, Grb10). How about other ICRs? It will be good to show DNA methylation results lacking Dnmt3a1 or Dnmt3a2 for most if not all of more than 20 known ICRs in mice.

We agree with the reviewer that a broader examination of ICRs is highly valuable. In addition to the data presented for *H19*, *Snrpn*, and *Grb10* in Fig. 5, our genome-wide EPIC array analysis enabled us to survey methylation states across numerous known ICRs. This approach identified significant hypomethylation at several other paternally methylated DMRs in the fertile *Dnmt3a2*^{-/-} sperm. We now include this new data in the Results section (lines 267-273) and as new Fig. 5E (see the figure on the right).

Minor points:

1. Some Dnmt3a may need to be either italic or capital letters.

We appreciate the reviewer for helping us improve the precision and clarity of our manuscript.

We have now corrected formatting for all genetic nomenclature.

2. It might be better to use “genomic imprinting” than “gene imprinting” in a few places. For example, Line 256.

We have now used the correct term of “genomic imprinting” in the revised manuscript in lines 264 and 274.

REVIEWERS' COMMENTS:

We thank both reviewers for their positive and constructive comments throughout the review process. We are pleased that the reviewers find the revised manuscript significantly improved and suitable for publication.

Please find our point-by-point response below:

Reviewer #1 (Remarks to the Author):

The revised manuscript demonstrates significant improvement through the inclusion of additional experiments and expanded discussion. While the DNA methylation analysis of germ cells from infertile *Dnmt3a2*^{-/-} males remains unresolved, I believe the current version is now suitable for publication in *Communications Biology*.

We thank the reviewer for the positive feedback and for acknowledging the significant improvements in our revised manuscript.

Reviewer #3 (Remarks to the Author):

The authors have properly addressed the concerns.

We thank the reviewer for the time and the positive comments on our revised manuscript.